# Strain Echocardiography to Predict Postoperative Atrial Fibrillation

**DOI:** 10.3390/ijms23031355

**Published:** 2022-01-25

**Authors:** Francisco Javier Sánchez, Esther Pueyo, Emiliano Raúl Diez

**Affiliations:** 1Faculty of Medical Sciences, National University of Cuyo, Mendoza 5500, Argentina; fsanchez@fcm.uncu.edu.ar; 2BSICOS Group, I3A, IIS Aragón, University of Zaragoza, 50018 Zaragoza, Spain; epueyo@unizar.es; 3CIBER-BBN, 28029 Madrid, Spain; 4Institute of Medical and Experimental Biology of Cuyo, IMBECU-UNCuyo-CONICET, Mendoza 5500, Argentina

**Keywords:** atrial fibrillation, cardiac surgery, pathophysiology, echocardiography, atrial strain

## Abstract

Postoperative atrial fibrillation (POAF) complicates 15% to 40% of cardiovascular surgeries. Its incidence progressively increases with aging, reaching 50% in octogenarians. This arrhythmia is usually transient but it increases the risk of embolic stroke, prolonged hospital stay, and cardiovascular mortality. Though many pathophysiological mechanisms are known, POAF prediction is still a hot topic of discussion. Doppler echocardiogram and, lately, strain echocardiography have shown significant capacity to predict POAF. Alterations in oxidative stress, calcium handling, mitochondrial dysfunction, inflammation, fibrosis, and tissue aging are among the mechanisms that predispose patients to the perfect “atrial storm”. Manifestations of these mechanisms have been related to enlarged atria and impaired function, which can be detected prior to surgery. Specific alterations in the atrial reservoir and pump function, as well as atrial dyssynchrony determined by echocardiographic atrial strain, can predict POAF and help to shed light on which patients could benefit from preventive therapy.

## 1. Introduction

Arrhythmias cause complications in 15% to 40% of patients during the postoperative period of cardiovascular surgery. The incidence of postoperative arrhythmias progressively increases from 18% in patients over 60 years old to 50% in octogenarians [1]. The pathophysiology of these arrhythmias is complex and involves a pre-existing and perioperative substrate.

Supraventricular tachyarrhythmias are the arrhythmias with the highest incidence. Atrial fibrillation (AF) is the most frequent arrhythmia after cardiovascular surgery in adults, representing approximately 80% of postoperative arrhythmias. AF incidence ranges from 15% to 40% for different centers around the world [2]. It generally occurs in the first five days after surgery, with the highest incidence found between the second and third days. The reversal rate to sinus rhythm is 80% while patients are in hospital and reaches 90% one month after surgery [3]. Despite reversibility, arrhythmias can cause angina pectoris, hypotension, and deterioration of the general state in the short term. The incidence of embolic stroke can increase the risk of cardiovascular mortality and hospital stay prolongation. In many cases, the use of antiarrhythmic drugs or electrical cardioversion is required for more effective treatment. When patients have sequelae due to arrhythmias, there is a remarkable increase in long-term health costs.

The main risk factors for postoperative AF (POAF) can be divided into preoperative, intraoperative and postoperative (Table 1). Older age, male sex, high blood pressure, diabetes mellitus, left ventricular systolic dysfunction, prolonged cross-clamp and pump time during coronary artery bypass graft procedures consistently raise the likelihood of POAF [4].

Several pathophysiological mechanisms of POAF have been elucidated in the last years. They cover a broad spectrum that includes oxidative stress, inflammation, alterations in calcium handling, atrial fibrosis, and altered mechanical function [5]. Still, with all this new knowledge, a clear strategy to prevent POAF is elusive. Therefore, the selection of patients that would benefit from a preventive approach in the perioperative setting is not well defined.

Many studies support the notion that POAF is a transient phenomenon, a “perfect storm” with severe consequences. This storm depends on the conjunction of several factors at the same time (inflammation, oxidative stress, autonomic imbalance, etc.), many of which are already present in the patient prior to surgery and could be detected by non-invasive methods. Preoperative echocardiography can help to predict this storm (Figure 1). Doppler echocardiography has shown specific capacity to detect patients prone to AF development based on the size of the atria and other Doppler parameters. Nevertheless, many patients escape this filter. Doppler echocardiography with strain technology has proved to be helpful in improving POAF prediction and the selection of patients that could benefit from a prevention strategy.

In this review, we aim to outline important mechanisms involved in POAF development in the cardiac surgery setting and to describe how strain echocardiography can recognize alterations in mechanical deformation of the atria related to proarrhythmic substrates, which could be used to optimize POAF prediction.

## 2. Mechanisms of POAF

### 2.1. Preoperative Substrate for the Development of POAF

Atrial remodeling refers to structural and functional changes in atrial myocardial tissue in response to various insults, promoting the development and perpetuation of AF. The most frequent causes of atrial remodeling include ventricular and valvular diseases, but, as described before, hypertension, diabetes, obesity, and old age are conditions that also contribute to atrial remodeling [5].

Aging is associated with significant atrial structural changes such as atrial fibrosis, augmented atrial size, and conduction abnormalities. There is evidence that age-related fibrosis is nonuniform but presents anisotropic distribution, which preferentially affects lateral rather than longitudinal conduction, thus promoting a slower and zig-zag-like propagation that increases the likelihood of reentry initiation by premature activations. Atrial connexin 43 (Cx43) downregulation in hearts of older humans has been associated with c-Jun N-terminal kinase activation, promoting reentry and AF in aged hearts [6]. Although it is well known that AF development increases with age, it is important to recognize that not all people age in the same way and individuals’ health status can range from fit to frail at all ages. Frailty is prevalent in AF patients [7].

The concept of “inflamm-aging” could also enlighten our understanding of the difference between aging and frailty. “Inflamm-aging” indicates an immune dysregulation in elderly people where increased levels of pro-inflammatory mediators accompany a blunted inflammatory response to immunogenic triggers [8]. Chronic elevations of IL-1, C-reactive protein, IL-6 and tumor necrosis factor are associated with frailty, motor and cognitive disability and the development of atrial remodeling and AF. Cellular senescence could also be related to AF because it shares multiple stimuli such as reactive oxygen species (ROS), telomere shortening, mitochondrial failure, and genomic damage [9].

Age-related electrophysiological, structural, and molecular changes in the atria cannot entirely explain the many faces of AF. An approach accounting for the comorbidities associated with age could better characterize the physiopathology and light the way for better prevention and treatment of AF.

Ischemia and atherosclerosis are preoperative diseases associated with oxidative stress and inflammation [10]. Increased nitrotyrosine and protein carbonyls formation in right atrial (RA) appendage biopsies suggest that oxidative stress plays an important role in POAF [5]. At this point, it is essential to notice that most of the analyses regarding atrial tissue alterations have been conducted using RA samples obtained during surgery before aortic cannulation. Inflammation and oxidative stress facilitate an arrhythmogenic substrate by several mechanisms. Tumor necrosis factor α, IL-6, and rennin-angiotensin system have been associated with AF [11]. A study in paroxysmal and chronic AF found that the most important source of superoxide in human atrial myocardium is NADPH oxidase [12]. NADPH oxidases are major sources of ROS in myocytes and vascular cells acting in response to cytokines and growth factors [13]. Patients with POAF have increased RA basal superoxide release, associated with higher tissue protein level of monoamine oxidase, NADPH oxidase 2, and p22phox subunits of NADPH oxidase. Although causality is yet to be demonstrated, atrial NADPH oxidase may be a key mediator of atrial oxidative stress that leads to the development of POAF [14]. Additionally, a study demonstrated the role of exogenous nitric oxide in reducing the incidence of POAF by using nitroprusside as a nitric oxide donor [15]. Furthermore, glutathione levels are low, and this seems to be associated with the downregulation of the L-type calcium current due to S-nitrosylation. Myocardial NADPH oxidase and dysfunctional NOS contribute to superoxide formation and oxidative damage in human atrial tissue in the setting of AF [16].

Other described alterations in patients who develop AF include modifications in Ca^2+^ handling by cardiac cells that promote ectopic firing. Oxidative stress plays a role in Ca^2+^-mediated triggers and AF initiation. This is caused by hyperphosphorylation of the ryanodine receptor (RYR2) via calmodulin-dependent protein kinase II (CaMKII), which can be activated by oxidation. Mitochondrial ROS production has been associated with ryanodine receptor oxidation that leads to increased intracellular Ca^2+^ leak and the development of AF [17,18]. This link is explained by the fact that Ca^2+^ released from the “leaky” RYR2 receptors in the sarcoplasmic reticulum would over-activate the Na^+^/Ca^2+^ exchanger to extrude Ca^2+^ and produce an arrhythmogenic depolarizing current, thereby explaining both the contractile dysfunction and the high recurrence rate of the arrhythmia [19].

An important aspect of structural remodeling is atrial fibrosis. The molecular basis of atrial fibrosis is poorly understood [5]. Left atrial (LA) fibrosis has been associated with increased expression of mRNAs encoding type I collagen and type III collagen and the profibrotic stimuli transforming growth factor-β and angiotensin II [20]. The angiotensin system is associated with markers of myeloperoxidase. Myeloperoxidase catalyzes the degeneration of hypochlorous acid, affecting intracellular signaling cascades in various cells and advancing the activation of metalloproteinases, ultimately resulting in the deposition of atrial collagen [21].

The substrates and mechanisms mentioned above are summarized in Table 2.

Angiotensin II activates the connective tissue growth factor by activating the small G protein Rac1 GTPase and NADPH oxidase. These contribute to signal transduction of structural remodeling in the left human atria, leading to up-regulation of Cx43, N-cadherin, and interstitial fibrosis. In other studies, atrial superoxide derived from NADPH oxidase and peroxynitrite have been associated with an increased risk of POAF [41].

It is important to note that a recent prospective study using RA appendages found that interstitial fibrosis and Cx43 distribution at the time of surgery were not associated with the incidence of POAF. Furthermore, there were no abnormalities with the levels of NT-proBNP, CRP, or oxidative stress biomarkers [42]. This shows the complexity of the pathophysiology of this arrhythmia and the still ongoing difficulties to better understand it, which is additionally hampered by limitations in the collection of LA samples.

In summary, these changes lead to structural remodeling and, therefore, contractile dysfunction of the atria, which play a fundamental role in the pathology of clinical AF and also POAF [43]. A summary of the described cellular and molecular mechanisms is shown in Figure 2.

### 2.2. Surgery Substrate for the Development of POAF

Surgery-related ischemia sensitizes the preoperative atrial substrate to postoperative AF triggers. Most atrial studies have used RA samples taken at the beginning of the surgery. Only a few studies have managed to collect atrial samples just before and after aortic cross-clamping, allowing signs of atrial ischemia-reperfusion to be analyzed [28,40,44].

Ischemia-reperfusion injury during cardiac surgery leads also to ROS formation, causing oxidative stress and systemic inflammatory response. At the myocyte level, this is accompanied by lipid peroxidation, cell membrane breakdown, decreased mitochondrial function, calcium overload, and apoptosis. This myocardial oxidative injury can increase susceptibility to POAF by impairing atrial contraction, altering myofibrillar energetics, and reducing the atrial effective refractory period [45]. The changes in ROS, inflammation, and autonomic dysfunction observed in human atrial samples during surgery promote both the formation of an arrhythmogenic substrate and ectopic firing (Table 2). Increased NFkβ protein levels and increased ROS production should prime and activate the NLRP3 inflammasome, which presents upregulated components in the atrial tissue of patients with POAF [46].

The mitochondrial function appears to be closely linked to arrhythmogenesis through ROS generation and the modulation of Ca^2+^ homeostasis [47]. A reduction in atrial mitochondrial respiration (oxidative transient postoperative substrate phosphorylation) has been identified in response to aortic cross-clamping [44]. In line with this, preoperative-to-postoperative differences in inflammatory markers, protein oxidation, and regulators of oxidative stress and atrial metabolism are more pronounced in patients with POAF than in those without POAF [28]. Some studies have associated impaired mitochondrial respiration with upregulation in the atrial expression of miR-1, miR-133a and miR-133b, although other studies have shown expression of miR-1 and miR-133a to be unaltered [40,44]. In RA biopsy samples taken before aortic cross-clamping and after reperfusion, caspase activity and the BAX:BCL2 mRNA ratio have been shown to increase and phosphorylation of BCL2 to decrease, suggesting promotion of apoptosis [28,40]. These changes were more pronounced in patients with POAF, suggesting that surgery-induced apoptosis might contribute to the development of POAF.

Interestingly, there is also evidence that, at the beginning of the postoperative period, there is only slight tissue damage and very little increase in the cardiac inflammatory response, with aortic cross-clamping not affecting the plasma levels of cardiac muscle troponin T, creatine kinase M-type/B-type (CK-MB), N-terminal pro-B-type natriuretic peptide, C-reactive protein or white blood cell count up to the first postoperative day [44]. Furthermore, total and phosphorylated RA protein levels of Cx40 and Cx43 are significantly reduced after aortic cross-clamping, even if there is no significant association with the incidence of POAF [48].

In paired RA samples obtained before venous cannulation and after myocardial reperfusion, atrial O_2_ production increases significantly because of increased mitochondrial and NOX2 activity [49]. The atrial content of antioxidant BH_4_ and nitric oxide synthase activity also decreases after reperfusion [49].

Even though the relationship between mitochondrial function, ROS production and POAF incidence has not been extensively studied, impaired mitochondrial function and increased ROS generation in response to ischemia-reperfusion are very likely to sensitize the atria to the development of POAF.

### 2.3. Postoperative Substrate for the Development of POAF

The postoperative atrial substrate is challenging to study. Studies in humans cannot be conducted due to ethical limitations. Therefore, animal models become crucial for understanding changes posterior to surgery. Animals undergoing surgery with pericardiotomy or right atriotomy (off-pump and on-pump surgery) show increased heterogeneity in atrial conduction and greater susceptibility to AF than control animals subject to anesthesia alone [50]. These changes are associated with increases in RA neutrophil infiltration along with increased myeloperoxidase activity. High-dose corticosteroid treatment of dogs subject to atriotomy suppresses POAF, suggesting that inflammation is an important contributor to AF in this model [50]. Furthermore, it was shown in these models that atrial incision is sufficient to cause inflammation and therefore changes in atrial electrophysiology that promote POAF.

Thus, inflammation appears to be important in the postoperative setting. The blood levels and atrial expression of IL-1β, IL-6, IL-17A, tumor necrosis factor (TNF), C-reactive protein, and TGFβ1 are significantly increased and are associated with epicardial thickening, leukocyte infiltration, and necrosis. Corticosteroids and statins prevent POAF in experimental sterile pericarditis, supporting the role of atrial inflammation [51,52,53].

Pericarditis-induced conduction slowing has been shown in several studies. This alteration might result from downregulation of Cx40 and Cx43 protein and altered physiological transmural gradients of connexin distribution. This promotes the formation of circuits that facilitate reentry [54,55].

Atrial fibrosis has also been described in this setting. Inhibition of pro-inflammatory cytokines, like IL 17A, decreases the atrial expression of IL-6, IL-1β and TGFβ, along with the levels of type I collagen, type III collagen, and α-smooth-muscle actin (αSMA), and increases the amount of MMP2 and MMP9, thereby reducing fibrosis and the duration of inducible AF episodes [56,57]. Cytokines might contribute to the development of AF by promoting cardiac inflammation and related atrial fibrosis.

## 3. Atrial Function Alteration: Strain Echocardiography as a Predictor of POAF

Since the late 1940s, cardiac ultrasonography has been a tool increasingly used to evaluate the heart and, in particular, the atria [58]. Bidimensional evaluation can measure the volume of the atria and determine its potential enlargement (anatomical evaluation). Doppler technology also allows the assessment of passive and active atrial function (functional evaluation) and the recognition of early signs of atrial remodeling. Atrial synchrony, size (total atrial volume), and function (pump, compliance) are all significant landmarks in the evaluation of the atria that can reveal pathological states (Figure 3). More recently, atrial synchrony has proved to be another useful tool for atrial functional assessment [14,59,60].

Doppler evaluation has been used to determine atrial function and pressure [61]. Transmitral inflow patterns by pulsed-wave Doppler sampling from the mitral leaflet tips provide insights into LA mechanical function. The peak transmitral A wave velocity is used as a measure of LA mechanical function. Furthermore, tissue doppler (TD) imaging can be used to determine left ventricular end pressures. The E/e’ relation determined by TD is a widely used marker of elevated end-diastolic pressure.

New echocardiographic measures are strain and strain rate, which represent the magnitude and rate of myocardial deformation. Speckle-tracking echocardiography (STE) is a reproducible, feasible, and easy-to-perform method to assess atrial function that overcomes several difficulties of previously used techniques and has stronger prognostic value [62]. STE assessment can be used to measure LA longitudinal strain, which is the most relevant parameter for functional analysis of the LA. STE allows the determination of phasic LA volumes, useful to evaluate atrial function.

Atrial deformation can be sub-divided into three phases:Reservoir phase: It starts at the end of ventricular diastole (mitral valve closure) and continues until mitral valve opening. It involves the time of left ventricular isovolumic contraction, ejection, and isovolumic relaxation. The atria fill with blood from the pulmonary veins or the cava veins during this phase.Conduit phase: It comprises from mitral valve opening through diastasis until the onset of LA contraction in patients in sinus rhythm.Contraction phase: It covers from the onset of LA contraction until the end of ventricular diastole (mitral valve closure) in patients in sinus rhythm [62].

Bidimensional echocardiogram with Doppler and Strain technology can therefore evaluate the structure and function of the beating heart and could show atrial hidden oxidation and inflammation, with signs of tissue remodeling potentially uncovered from analysis of atrial dynamic cyclic changes. This makes this image tool useful for the prediction of POAF. Peak transmitral A wave velocity has been shown to be absent in AF, even if this alteration has not been proven to be an independent marker of POAF. The E/e’ relation has been correlated with larger atria and the development of POAF [61]. Other studies have associated prolonged atrial conduction time measured by TD with atrial electrical remodeling resulting in POAF [63]. TD has also been linked to RA fibrosis in samples obtained before surgery [64]. Atrial dyssynchrony determined by TD has been used to predict POAF [65].

The relationship between echocardiographic alterations and the incidence of POAF has been recently reviewed by Kawczynski et al. [66]. Table 3 summarizes the main results regarding echocardiographic atrial strain and POAF. Altered LA reservoir strain is associated with fibrosis in the atria and may predict the risk of new-onset AF [30,67,68,69]. In severe aortic stenosis, impaired LA reservoir and booster pump functions predicted the occurrence of new-onset AF after aortic valve replacement, independently of LA enlargement [1,69,70,71,72,73]. In patients with severe mitral disease, reservoir and contractile LA alterations measured with STE were associated with the development of POAF [69,74,75]. Right atrial enlargement and alteration of function determined by longitudinal STE have been also associated with the development of POAF [76]. 

Disturbances in the timing of LA contraction reflect the presence of atrial fibrosis and electrophysiological disorders. Indeed, intra-atrial dyssynchrony during sinus rhythm is an independent predictor of recurrence after the first AF ablation [59]. LA dispersion assessed by STE has been shown to be an independent predictor of new-onset AF [77].

## 4. Electrical and Structural Changes in Atrial Tissue Associated with Echocardiographic Findings in the Preoperative Setting

Numerous studies show that subclinical LA booster pump, conduit and reservoir function alterations determined by atrial strain are useful tools for risk stratification and prevention of POAF. A recent meta-analysis found that increased atrial size and reduced reservoir atrial function have a strong correlation with POAF incidence [66].

Several lines of evidence also suggest that intra-atrial dyssynchrony determined by echocardiography contributes indispensably to AF mechanisms and the prediction of AF incidence in different clinical scenarios, including the perioperative period of cardiovascular surgery. This can be associated with LA electrical remodeling independent of LA volume and with fibrosis burden not always detected by cardiac magnetic resonance [59,78,79].

Table 4 highlights studies relating reservoir and pump function with structural and electrical alterations [26,68,80,81].

LA enlargement has been associated with the development of AF in the postoperative setting [82]. Larger atrial sizes are associated with higher levels of fibrosis, which alters atrial structure and function. It is common to find a high correlation between P wave duration in the ECG and LA size. Nonetheless, some clinical studies have shown that P wave duration and atrial size are not necessarily linked. In these cases, the presence of Bayes syndrome (interatrial block) could be only an electrical phenomenon. The causes and mechanisms of interatrial block exceed the scope of this review, but it is important to notice that this atrial conduction alteration has been associated with AF, although the data in the cardiovascular surgery setting are scarce and contradictory [83,84,85,86].

Gap junction remodeling, connexin dysregulation, and atrial conduction slowing play an important role in this electrical remodeling [87]. Atrial gap junctions consist of connexin proteins, mainly Cx40 and Cx43. Oxidative stress has been shown to alter the atrial expression of Cx40 and Cx43 as well as the size of atrial gap junctions in a model of obstructive sleep apnea. It also disrupts Cx43 forward trafficking to the intercalated disk resulting in abnormal gap junction coupling [88]. Cx lateralization is associated with interatrial dyssynchrony measured by strain echocardiography [14].

Oxidative stress is associated with elevated ROS, which can be increased by directly affecting Ca^2+^ or indirectly through the cell membrane to produce intracellular Ca^2+^ overload. There are many different Ca^2+^ channel proteins in cardiac muscle cells. The Ca^2+^ channel in the sarcolemma is an important component of the action potential and the excitation function of myocardial cells. Reduced SERCA activity is an essential cause of impaired excitation–contraction coupling and contributes to the arrhythmogenic substrate, with LA longitudinal pump strain being significantly reduced [17,18,26].

In summary, the echocardiogram is currently a fundamental non-invasive tool for the analysis of the atria in the preoperative setting of cardiovascular surgery. The use of known parameters such as indexed atrial volume and end-diastolic pressures by tissue Doppler are sensitive markers for the occurrence of POAF. More recently, the use of strain echocardiography, with the determination of the different phases of atrial function (reservoir, conduit, and contraction), allows more sensitive assessment of their function and the early detection of alterations that would facilitate the appearance of POAF.

## 5. Conclusions

POAF is a frequent complication after cardiac surgery that can affect a patient’s outcome. The evidence reviewed here shows that the combination of pre-existing factors, such as previous diseases and frailty, set the stage for the remodeling of the atria, which, in combination with intraoperative and postoperative factors, promote POAF initiation. This cascade of events is mediated by atrial oxidative stress, inflammation, fibrosis, apoptosis and electrical remodeling. These pathophysiological changes set the stage for alterations in atrial function. A preoperative echocardiogram with strain technology can reveal these alterations by assessing a large set of variables in an effective and non-invasive way. A better understanding of atrial dysfunction may contribute to prediction of atrial fibrillation in the postoperative setting.

## Figures and Tables

**Figure 1 ijms-23-01355-f001:**
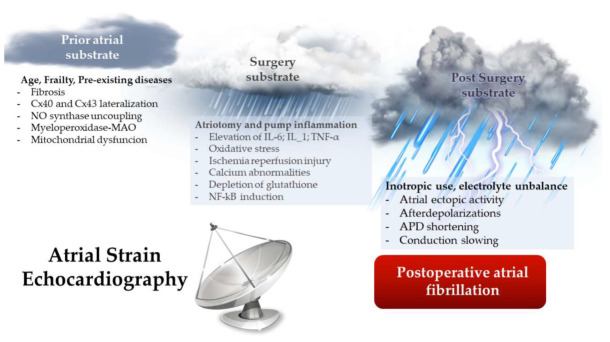
“The perfect storm”. Mechanisms involved in the development of atrial structural and electrical remodeling that facilitate the incidence of POAF. APD: action potential duration. Cx: connexin. ROS: reactive oxygen species. NO: nitric oxide. MAO: monoamine oxidase. IL: interleukin. NF-kB: nuclear factor-kappa B.

**Figure 2 ijms-23-01355-f002:**
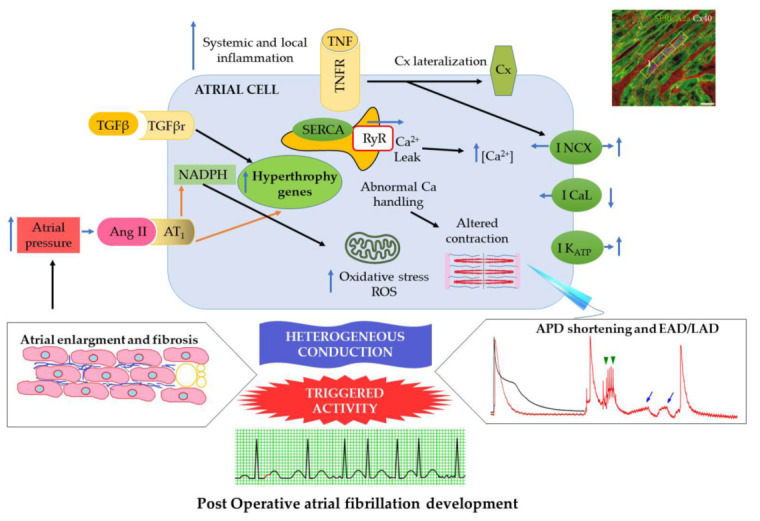
Summary of mechanisms related to atrial electrical and structural remodeling described in POAF. Ang II: angiotensin II. APD: action potential duration. Cx: connexin. EAD (green arrowheads): early afterdepolarizations. LAD (blue arrows): late afterdepolarizations. NCX: sodium–calcium exchanger. ROS: reactive oxygen species. TNF: tumor necrosis factor. TNFR: tumor necrosis factor receptor. NADPH: nicotinamide adenine dinucleotide phosphate. TGFβ: transforming growth factor-beta. TGFβr: transforming growth factor-beta receptor.

**Figure 3 ijms-23-01355-f003:**
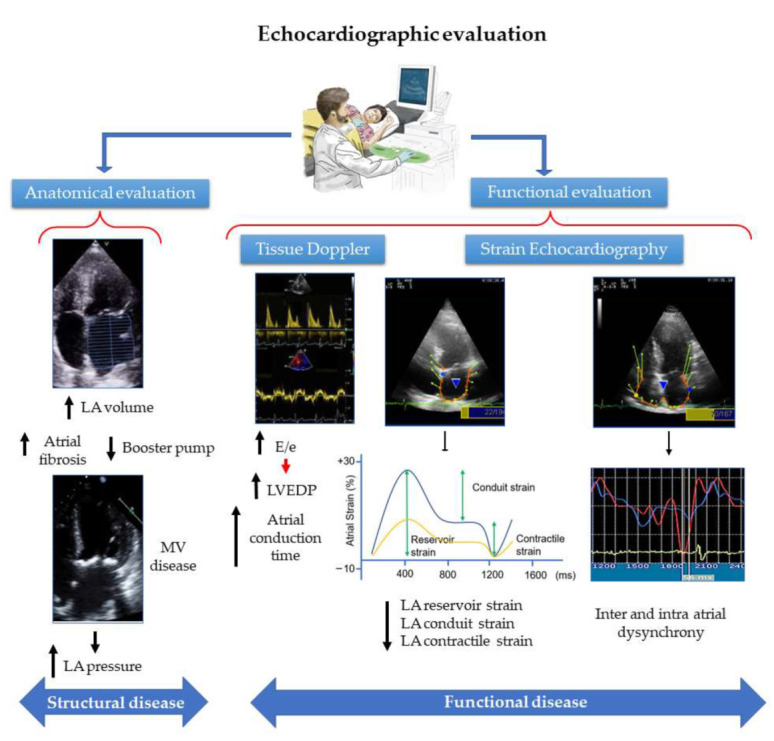
Echocardiographic evaluation of the atria. Anatomical evaluation is based mainly on atrial volume and also on atrioventricular valves disease. Functional evaluation of the atria allows measuring left ventricular end-diastolic pressure (LVEDP) and total atrial conduction time, both related to POAF. Strain echocardiography enables assessment of reservoir, conduit, and longitudinal contractile strain. It also allows intra- and inter-atrial dyssynchrony evaluation. LA: left atria. MV: mitral valve. E/e: relation between early diastolic phase in pulsed Doppler and tissue Doppler. Atrial strain %: Blue line: normal longitudinal atrial strain. Yellow line: impaired atrial longitudinal strain.

**Table 1 ijms-23-01355-t001:** Perioperative clinical risk factors associated with the risk of POAF.

Preoperative Risk Factors	Intra-Operative Risk Factors	Postoperative Risk Factors
Advanced age	Aortic cross-clamp time	Respiratory compromise
Male sex	Type of surgery	Red cell transfusion
Hypertension	On-pump time	Use of adrenergic drugs
COPD	Bicaval canulation	
Heart failure	Systemic hypothermia	
Left ventricular hypertrophy		
Renal failure		
Obesity		
Withdrawal of beta-blockers		
Diabetes mellitus		

POAF: postoperative atrial fibrillation. COPD: chronic obstructive pulmonary disease.

**Table 2 ijms-23-01355-t002:** Substrates for POAF from human atrial samples obtained at the start of surgery.

Substrate	Tissue	Molecular/Function	References
Electrophysiology	↔/↓ dV/dtmax	↔ SCN5A	[14,22]
	↔ APD/ERP ↓ APD	↔ I_Ca,L_ ↔ I_K1_ ↔ I_K,Ach_ ↔ I_Kur_ ↔ I_to_, ↑ I_KATP_	[14,22,23,24,25]
	calcium handling	↓ sarcolipin ↔ SERCA2a ↔ phospholamban ↔ RYR2 ↔ IP3R ↔ Na^+^/Ca^2+^ exchanger	[23,26,27]
Structural remodeling	↔/↑ RA fibrosis ↑ LA fibrosis	↑ TGFβ1 ↔/↑ type I and type III collagen	[20,28] [24,29,30]
	↔/↑ apoptosis ↑ myocytolysis	↑ Apoptosis-inducing factor ↔ p-BCL-2 ↔ Caspase 3	[31]
	↑ Myocyte hypertrophy	↓ Myosin 6 ↑ Myosin 7	[31]
		↔ β1-AR, ↔ β2-AR	[23]
Cell–cell coupling	Connexin 40 lateralization	↔/↑ Connexin 40	[14]
	Connexin 43 lateralization	↔ Connexin 43	[14]
		↔ Connexin 45	[32]
		↓ Connexin 40:connexin 43 ratio	[33]
Oxidative stress		↑ ROS ↓ Peroxiredoxin 1 ↑ NOX2 ↑ p22_phox_ ↔ MnSOD	[34,35,36]
	↑ 3-nitrotyrosine	Peroxinitrites	[14]
Inflammation	↑ TNFα	↑/↓ NF-κB ↔/↑ IL-6 ↑ TLR4 ↑ NLRP3 ↑transferrin	[34,35,37,38]
MicroRNAs		↓ MicroRNA-195 ↓ MicroRNA-199a ↔ MicroRNA-1 ↔ MicroRNA-133a	[39,40]

ADP: action potential duration. AR: adrenergic receptor. SCN5A: sodium voltage-gated channel alpha subunit 5. dV/dtmax: maximum upstroke rate of the action potential. ERP: effective refractory period. LA: left atria. RA: right atria. SERCA2A: sarcoplasmic reticulum Ca^2+^-ATPase 2a. I_CaL_: L-type Ca^2+^ current. I_f_: hyperpolarization-activated (funny current). I_K1_: inward-rectifier K^+^ current. I_K,ACh_: acetylcholine-activated inward-rectifier K^+^ current. I_Kr_: rapid delayed-rectifier K^+^ current. I_Kur_: ultrarapid delayed-rectifier K^+^ current. I_Na_: Na^+^ current. IL-6: interleukin 6. IP3R: inositol 1,4,5-trisphosphate receptor. I_to_: transient outward K^+^ current. MnSOD: manganese superoxide dismutase. NF-κB: Nuclear factor κB. NOX2: NADPH oxidase 2. ROS: reactive oxygen species. NLRP3: NOD-, LRR- and pyrin domain-containing protein 3. RYR2: ryanodine receptor 2. TGFβ1: transforming growth factor β1. p-BCL-2: phosphorylation of B cell lymphoma 2. TLR4: toll-like receptor 4.

**Table 3 ijms-23-01355-t003:** Echocardiographic variables in strain analysis associated with the risk of POAF.

Study (Year, Type Surgery)	N° Patients	POAF (%)	Results
Tayyareci et al. (2010, CABG)	96	26	LA reservoir strain < 44% predicted POAF (Se: 88.7%; Sp: 96%; *p*: 0.0001)
			LA systolic strain rate < 1.7 s^−1^ predicted POAF (Se: 88%; Sp: 86.2%; *p*: 0.0001)
			LA conduit strain rate < 1.95 s^−1^ predicted POAF (Se: 72%; Sp: 70.4% *p*: 0.0001)
Gabrielli et al. (2011, CABG)	70	26	LA contractile strain rate impairment predicted POAF (*p*: <0.01)
			LA reservoir strain rate impairment predicted POAF (*p*: <0.01)
Her et al. (2013, CABG)	53	24	LA reservoir global strain < 27.7% predicted POAF (Se: 81%; Sp: 69%; *p*: <0.003)
Imanishi et al. (2014, AVR)	27	56	LA contractile strain rate > 0.79 s^−1^ predicted POAF (Se: 60%; Sp: 92%; *p*: <0.0001)
Cameli et al. (2014, AVR)	76	19.7	LA reservoir global strain < 16.8% predicted POAF (*p*: <0.0001; HR 6.55; Se: 86%; Sp: 91%)
Verdejo et al. (2016, CABG)	70	38.5	LA reservoir global strain impairment predicted POAF (*p*: <0.001)
Basaran et al. (2016, CABG)	90	25.6	LA reservoir impairment predicted POAF (*p*: <0.0001)
Pernigo et al. (2017, AVR)	60	43.3	LA strain reservoir < 23% predicted POAF (*p*: 0.0001)
			Atrial strain before contraction < 10% predicted POAF (*p*: 0.0001)
Sabry et al. (2017, MVR)	50	44	LA reservoir strain < 23% predicted POAF (Se: 90.9%; Sp: 93.33%; *p*: 0.003)
Pessoa-Amorin et al. (2017, MVR)	115	36.7	LA strain reservoir < 18.7% predicted POAF independent of atrial volume (*p*: 0.039)
			Atrial strain before contraction < 7.9% predicted POAF (*p*: 0.038)
Lisi et al. (2018, MVR)	36	32	Lower values of LA reservoir strain were associated with POAF (*p*: 0.0001)
Aksu et al. (2019, CABG)	74	50	RA reservoir strain < 11 predicted POAD (Se: 72%; Sp: 65% *p*: 0.001)

POAF: postoperative atrial fibrillation. LA: left atria. RA: right atria. HR: hazard ratio. CABG: coronary artery bypass grafting. MVR: mitral valve replacement. AVR: aortic valve replacement. Se: sensibility. Sp: specificity.

**Table 4 ijms-23-01355-t004:** Preoperative echocardiographic findings and their tissue alteration correlation.

Study (Year)	N° Patients	Endpoint	*p* Value	Echocardiographic Alteration	Substrate Alteration
Sanchez et al. (2020)	45	POAF	0.0416	Interatrial dyssyncrhony	AP shortening, Cx40 lateralization, higher nitrotyrosine signal, K_ATP_ increased signal
Fakuade et al. (2020)	202	POAF	<0.05/<0.01	Reduced pump and reservoir function	Reduction of SR Ca^2+^ release in atrial myocytes.
Mandoli et al. (2020)	65	HF and mortality	0.0001	Reservoir LA strain	Atrial fibrosis
Gasparovic et al. (2014)	44	Effect of AR on atrial strain rate	0.006/0.001	Strain rate reservoir and pump atrial function	Atrial fibrosis and apoptosis

AP: action potential. Cx: connexin. POAF: postoperative atrial fibrillation. LA: left atria. HF: heart failure. AR atrial remodeling. SR: sarcoplasmic reticulum.

## Data Availability

Not applicable.

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
