# Peer review of "Strain Echocardiography to Predict Postoperative Atrial Fibrillation"

_ijms, 2022, doi:10.3390/ijms23031355_

Round 1
Reviewer 1 Report
Thank you for allowing me to review this work. The paper is generally well written and the references are up-to-date.
I have only minor suggestion:
- English form is fair, but should be improved for an international audience
- In the Introduction, I suggest adding a brief sentence describing, if present, the short-term (i.e. hypotension) and long-term (i.e. stroke/thromboembolism) impact of POAF, and its therapeutic management during and after admission.
Author Response
Authors' Responses to Reviewers Comments (Reviewer 1)
Author’s Notes:
“English form is fair, but should be improved for an international audience. In the introduction, I suggest adding a brief sentence describing, if present, the short-term (i.e. hypotension) and long-term (i.e. stroke/thromboembolism) impact of POAF, and its therapeutic management during and after admission.”
Answer :
As suggested by the reviewer, several grammatical and stylistic modifications were implemented.
Also, the following lines have been added to the introduction: “Despite reversibility, arrhythmias can cause angina pectoris, hypotension, and deterioration of the general state in the short term. The incidence of embolic stroke can increase the risk of cardiovascular mortality and hospital stay. In many cases, it is mandatory to use antiarrhythmic drugs or electric cardioversion for more effective treatment. It is also important to mention the increase in long-term health costs when patients have sequelae due to arrhythmias.”
Reviewer 2 Report
The manuscript in reference describes some mechanisms involved in postoperative atrial fibrillation (POAF) development and some parameter to optimize POAF prediction through strain echocardiography details and related alterations. The manuscript is interesting and contains relevant information for readers. However, there are some particular points to be addressed by the authors.
Detailed scrutiny should be done throughout the manuscript to correct some stylistic and grammar issues.
The explanation/description and discussion of Figure 2 must be expanded since it is not totally linked with the information in section 2. This comment also applies to Figures 1 and 3.
Authors can improve the order of ideas of several passages since such ideas are located without connectivity and make it difficult to understand and track the complete information. Furthermore, some paragraphs are extensive, and others are very short or summarized, and a reason is not entirely logical.
The authors mentioned that the prediction of atrial fibrillation can be performed through preoperative echocardiography under the understanding of atrial dysfunction. However, this critical part is not totally clear in the manuscript. I suggest including a final section, previous conclusions, with the important, effective set of variables to predict POAF.
Author Response
Authors' Responses to Reviewers Comments (Reviewer 2)
Author's Notes
The manuscript in reference describes some mechanisms involved in postoperative atrial fibrillation (POAF) development and some parameter to optimize POAF prediction through strain echocardiography details and related alterations. The manuscript is interesting and contains relevant information for readers. However, there are some particular points to be addressed by the authors.
- Detailed scrutiny should be done throughout the manuscript to correct some stylistic and grammar issues.
Answer: Thank you very much for the suggestions. All authors made a detailed manuscript revision and implemented substantial stylistic and grammatical modifications.
- The explanation/description and discussion of Figure 2 must be expanded since it is not totally linked with the information in section 2. This comment also applies to Figures 1 and 3.
Answer: The discussion of Figure 2 was expanded and mentioned in the text to extend its use to the reader. The same was done with figures 1 and 3.
- Authors can improve the order of ideas of several passages since such ideas are located without connectivity and make it difficult to understand and track the complete information. Furthermore, some paragraphs are extensive, and others are very short or summarized, and a reason is not entirely logical.
Answer: The article was ordered to make it more understandable. All the sections describing mechanisms were joined. Echocardiogram sections were divided to clarify the clinical results from those to describe the relation to tissue alterations. All these changes intend to make the text more coherent.
- The authors mentioned that the prediction of atrial fibrillation can be performed through preoperative echocardiography under the understanding of atrial dysfunction. However, this critical part is not totally clear in the manuscript. I suggest including a final section, previous conclusions, with the important, effective set of variables to predict POAF.
Answer: Thanks for your valuable suggestion. As requested, the following lines have been added to section 4 of the review “In summary, the echocardiogram is currently a fundamental non-invasive tool for the analysis of the atria in the preoperative setting of cardiovascular surgery. The use of known parameters such as indexed atrial volume and end-diastolic pressures by tissue Doppler are sensitive markers for the occurrence of POAF. More recently, the use of strain echocardiography, with the determination of the different phases of atrial function (reservoir, conduit, and contraction), allows a more sensitive assessment of their function and the early detection of alterations that would facilitate the appearance of POAF. “